# Uniform Distribution of Pd on GO-C Catalysts for Enhancing the Performance of Air Cathode Microbial Fuel Cell

**Jingzhen Wang [1,2], Kaijie Mu [2], Xuedong Zhao [2], Dianliang Luo [2], Xiaodi Yu [2], Wenpeng Li [3,4,*] , Jie Chu [5], Jing Yang [6,*] and Qinzheng Yang [1,2,*]**

1  State Key Laboratory of Biobased Material and Green Papermaking, Qilu University of Technology, Shandong Academy of Sciences, Jinan 250353, China; wjz_424@163.com
2  Department of Bioengineering, Qilu University of Technology, Shandong Academy of Sciences, Jinan 250353, China; mukaijie@126.com (K.M.); zhaosimon666@yeah.net (X.Z.); luodianliangldl@163.com (D.L.); xiaodiyu1996@126.com (X.Y.)
3  Engineering & Technology Center of Electrochemistry, School of Chemistry and Chemical Engineering, Qilu University of Technology, Shandong Academy of Sciences, Jinan 250353, China
4  Shandong Key Laboratory of Biochemical Analysis, College of Chemistry and Molecular Engineering, Qingdao University of Science and Technology, Qingdao 266042, China
5  Biology Institute, Qilu University of Technology, Shandong Academy of Sciences, Jinan 250103, China; chujie6532@163.com
6  Department of Physics, School of Electronic and Information Engineering, Qilu University of Technology, Shandong Academy of Sciences, Jinan 250353, China
*  Correspondence: liwenpeng@qlu.edu.cn (W.L.); yangjing@qlu.edu.cn (J.Y.); yqz@qlu.edu.cn (Q.Y.)

**Abstract:** Metal, as a high-performance electrode catalyst, is a research hotspot in the construction of a high-performance microbial fuel cell (MFC). However, metal catalyst nanoparticles and their dispersed carriers are prone to aggregation, producing catalytic electrodes with inferior qualities. In this study, Pd is uniformly dispersed on the graphene framework supported by carbon black to form nanocomposite catalysts (Pd/GO-C catalysts). The effect of the palladium loading amount in the catalyst on the catalytic performance of the air cathode was further studied. The optimized metal loading afforded a reduced resistance and improved accessibility of Pd particles for the ORR. The maximum current output of the 0.250 Pd (mg/cm$^2$) MFC was 1645 mA/m$^2$, which is 4.2-fold higher than that of the carbon paper cathode. Overall, our findings provide a novel protocol for the preparation of high-efficient ORR catalyst for MFCs.

**Keywords:** oxygen reduction reaction; microbial fuel cell; air cathode

## 1. Introduction

A microbial fuel cell is a renewable energy device that uses bacteria as a biocatalyst to convert the chemical energy into electrical energy via biochemical pathways during the wastewater treatment process [1–4]. Microbial fuel cell incorporating air-breathing cathodes is a new type of MFC (Figure 1), which had various advantages such as low operating cost and zero secondary pollution [2,5]. In an air cathode MFC, electrons produced by the bacteria are transferred to the anode and flow to the cathode. In most cases, oxygen in the air is used as a sustainable oxidizer at the cathode [6]. The oxygen enters the catalytic layer and reacts with the electrons and protons flowing into the cathode to form water under the action of the cathode catalyst [7,8]. However, currently, the development of more effective and stable catalysts is highly desired for the cathodic oxygen reduction reaction (ORR) in MFCs [9–13].

Fast ORR can be triggered effectively with precious metals [14–17], since they form coordination bonds and participate in ORR through their vacant d orbits [14,18]. The lack of uniformity in the distribution of precious metals often causes the burial of active sites, resulting in low catalytic efficiency. Therefore, many researchers began to use some precious metals as catalysts dispersed on nano-carbon-based supports to solve the issue of low

catalytic efficiency [19]. Graphene contains a single-atom thick and two-dimensional structure, and it is a superior material to enhance the interaction and dispersion of metal [20]. Therefore, graphene could improve the redox reaction [18,21,22], which has been previously demonstrated by the study of Pd catalysts supported on other types of carbon materials [23–27].

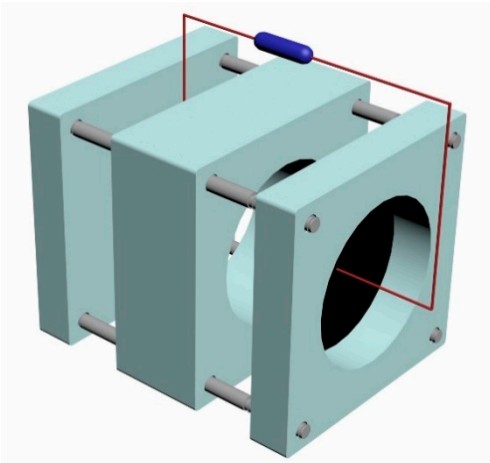

**Figure 1.** Microbial fuel cell operating with an air cathode.

However, graphene nanosheets with precious metals easily agglomerate together during the preparation of catalysts. The agglomeration of graphene can be avoided effectively by adjusting the layer distance of graphene. Many methods have been used to solve this problem, such as adding carbon black and carbon tubes in graphene materials to increase the layer distance and prevent graphene agglomeration [28,29]. Since a uniform distribution of precious metal particles on the surface of carbon material is critical to electrocatalysis, the distribution of metal particles is a crucial step.

In this study, after the carbon black and graphene were mixed to construct a skeleton structure, palladium was uniformly dispersed in the graphene/carbon black composite structure. The catalyst performance was optimized by adjusting the loading of the catalyst in the cathode. X-ray photoelectron spectroscopy (XPS), transmission electron microscopy (TEM), scanning electron microscopy (SEM), and energy-dispersive spectrometer (EDS) were used to investigate the structure and morphology characterization of cathode catalysts. The electrochemical activities of the electrodes were investigated by measuring their cyclic voltammograms (CV) and electrochemical impedance spectroscopy (EIS). Finally, we developed a process for dispersing Pd and its applications, providing a valuable reference for the development of air cathode (AC).

## 2. Results and Discussion

### 2.1. Characterization of Pd/GO-C Catalyst

In this study, we firstly mixed the GO, carbon black, $PdCl_2$, and reducing agent $NaBH_4$ in an aqueous solution, allowing the GO to be well dispersed with carbon black. At the same time, $Pd^{2+}$ was reduced to Pd by 5% $NaBH_4$, which attached to the mixture of GO and carbon black to afford a Pd/GO-C complex catalyst (Figure 2). The elemental composition of catalyst mixture and chemical valence were investigated by X-ray photoelectron spectroscopy (XPS). The element composition from XPS spectra is shown in Figure 3a, in which two peaks showed the presence of C (284 eV) and Pd (338 eV), respectively (NIST X-ray Photoelectron Spectroscopy Database, Version 4.1, USA; National Institute of Standards and Technology, Gaithersburg, 2012; http://srdata.nist.gov/xps, accessed on 6 July 2020). The high resolution of the Pd 3d spectrum represented two pairs of peaks (Figure 3b), indicating the existence of two different Pd oxidation states on the surface. The relative lower binding energy set of two peaks (334.2 and 339.8 eV) was ascribed to 0 valent Pd, whereas

the other set of two peaks at higher binding energies (335.9 and 341.2 eV) was assigned to the +2-oxidation state of Pd ($Pd^{2+}$). The existence of $Pd^{2+}$ indicated that Pd nanoparticle was dispersed uniformly in the catalyst, which was attributed to the incorporation of the 3D structure of GO and carbon black. As a result of the addition of carbon black in the mixing process, the carbon black enters into the GO layer, which prevents the aggregation of GO and ensures the uniform dispersion of Pd between GO lamellae [19]. The uniform dispersion of Pd atoms could be readily oxidized to $Pd^{2+}$ by oxygen in the air. The oxidized Pd can also play a positive role in the catalytic reduction of oxygen as a Pd atom [30–32].

The dispersion of palladium, GO, and carbon black in the catalyst mixture plays an important role in the catalysis activity of catalyst. To manifest the structure change, the surface morphological structure of the Pd/GO-C catalyst mixture dry powder was investigated by SEM before being coated on the carbon paper (Figure 3c). At the same time, we analyzed the surface morphology and elemental composition of catalyst using EDS. According to the SEM result (Figure 3d), the presence of Pd on the electrode was confirmed by EDS analysis (Figure 3e), and the energy spectrum of the element ratio was consistent with previous results. Figure 3f shows the mapping analysis of Pd elements, which confirms that Pd is uniformly dispersed during the loading process. At the same time, through the determination of EDS, the element concentration of C, N, and Pd were obtained, which were 44.25%, 34.51%, and 21.24%. TEM and HR-TEM were also used to identify the catalyst, and the results further confirmed the uniform distribution of palladium in the catalyst (Figure S1a,b). The uniform distribution could ensure the high catalytic performance of catalysts when used as an air cathode.

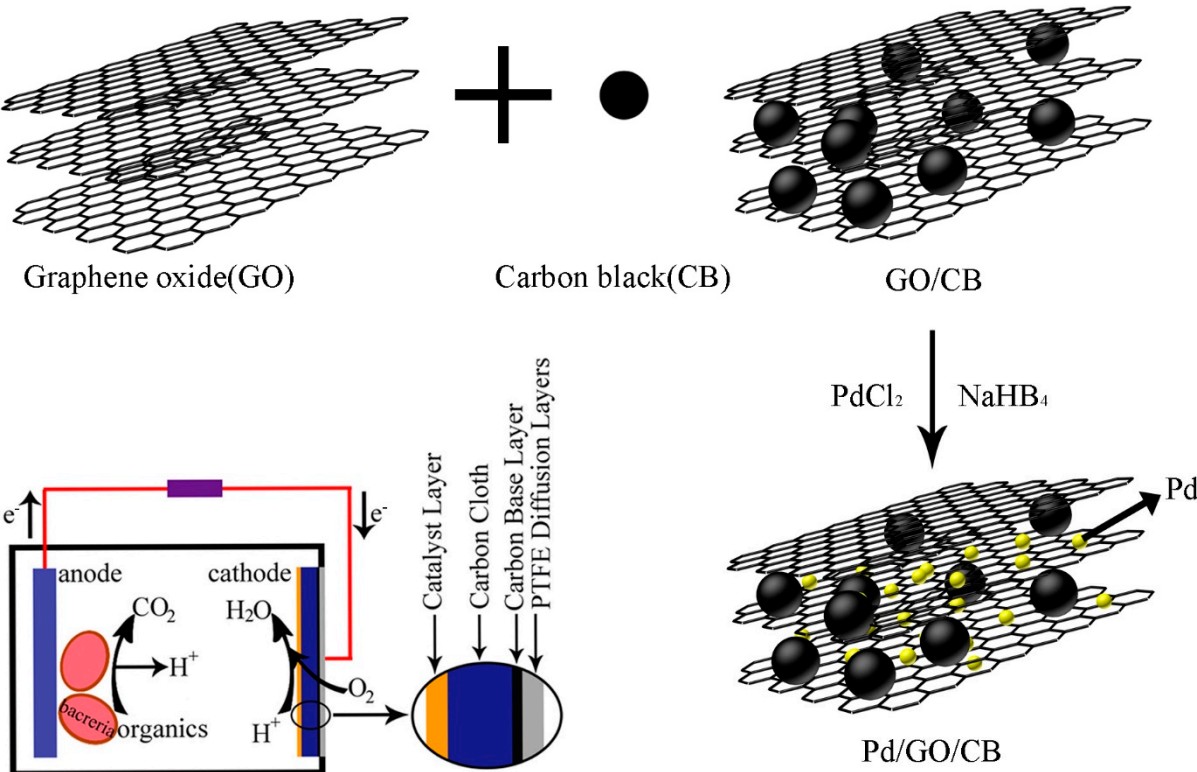

**Figure 2.** Schematic illustration of the fabrication process of the Pd/GO-C cathode catalyst.

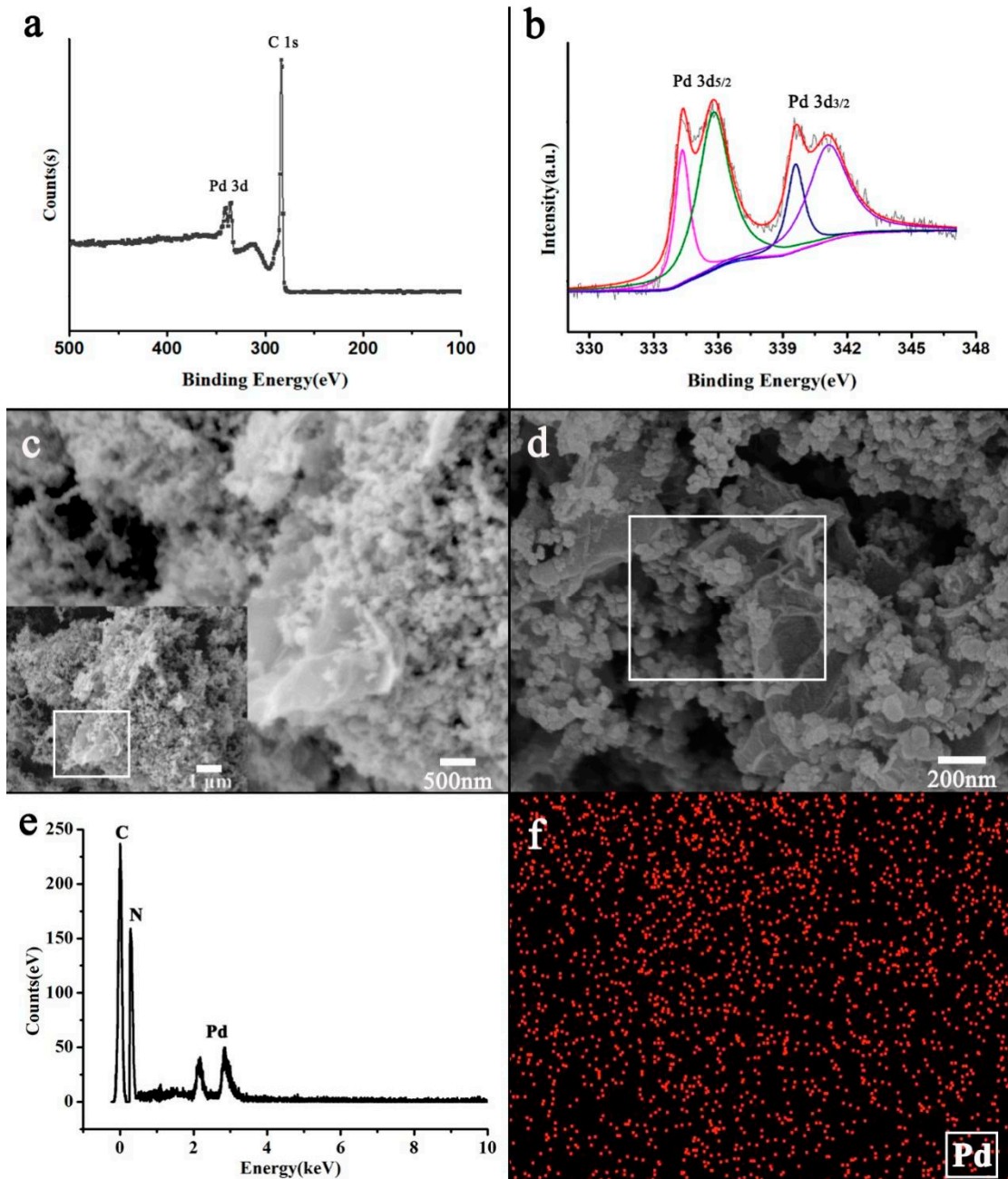

**Figure 3.** (**a**) XPS image of cathode mixture demonstrated the presence of Pd and C. (**b**) XPS spectra of Pd 3d5/2 and 3d3/2 binding energy regions for Pd/GO-C. (**c**) SEM images for Pd/GO-C mixture. (**d**,**e**) EDS analysis of catalyst. (**f**) Mapping analysis of Pd elements.

Our results showed that bulky graphite materials expanded to an apparent layer structure upon chemical oxidation. Carbon black and GO are two kinds of nanomaterials with different dimensions. They form a 3D structure that can inhibit the agglomeration of graphene lamellae, resulting in a synergistic effect and providing a very favorable structural condition for the functional design of nanocomposites. These complex structures can increase the contact area between the cathode and air, thus improving the cathode performance. The existence of palladium was confirmed by XPS and EDS. The results of mapping analysis and TEM indicated that the distribution of Pd in the catalyst was uniform. The GO is favorable for fixing metals, and the carbon black can prevent the

stacking of graphene. Since both surfaces of graphene can contact Pd, the conductivity of the cathode is significantly enhanced.

By employing this method, we applied the prepared Pd/GO-C catalyst mixture to the surface of the carbon paper to obtain the Pd/GO-C cathode and employed these electrodes with a rich pore structure to enhance the performance of the MFC.

### 2.2. Performance of MFC with Catalysts in the Cathode

To elucidate the role of catalyst, the cathode catalyst layers with 0.125, 0.250, and 0.500 mg/cm$^2$ palladium loading were employed to verify their performance of electricity generation in MFCs, and carbon paper without catalyst was used as the blank control [33,34]. Significant differences in the performance of three different Pd/GO-C were observed. The current density of the cathode with Pd was higher than that of control. The current density of the cathode loaded with 0.250 Pd was about 25% higher than those with 0.125 Pd and 0.500 Pd (Figure 4a). Compared with the Pt/C cathode, the current density of 0.250 Pd was slightly less than that of the platinum cathode. During acclimation of the reactors using the cathodes loaded with 0.250 Pd, the current produced by the MFCs (1 KΩ) increased from 1498/mAm$^{-2}$ to 1645/mAm$^{-2}$ after only three cycles of operation (Figure 4b). Polarization and power density curves obtained by varying the external loading resistances are used as another standard protocol to evaluate MFC performances (Figure 4c). High maximum power densities were based on the low internal resistance. The maximum power density of the 0.250 Pd-MFC was 901 mW/m$^2$, which was 1.6-fold and 1.8-fold higher than those of the 0.500 Pd-MFC (559 mW/m$^2$) and 0.125 Pd-MFC (502 mW/m$^2$), respectively. Compared with Pt/C (925 mW/m$^2$), the power density of 0.250 Pd-MFC was slightly less, indicating that the performance of 0.250 Pd-MFC is similar to that of Pt/C MFC.

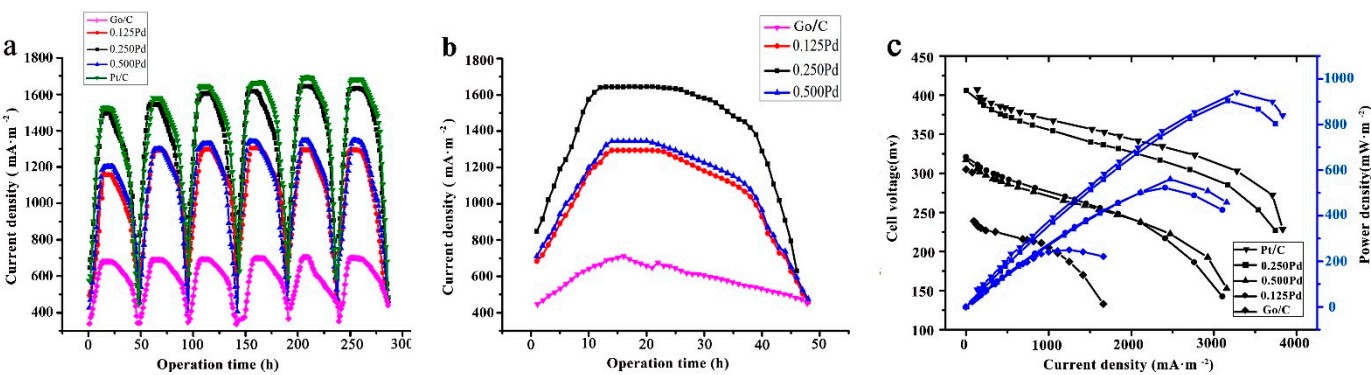

**Figure 4.** (**a**,**b**) The current density–time graph of the MFCs equipped with different cathodes. (**c**) Polarization curves and power outputs of the MFCs.

The MFC with Pd loading of 0.250 mg/cm$^2$ as cathode catalyst exhibits the best performance because of its highest maximum power density and the lowest internal resistance. Therefore, the excellent ORR activity of the Pd/GO-C cathode is mainly attributed to two factors. Firstly, spherical C particles accumulate on the wrinkled surface of the GO to form a GO-C hybrid with a 3D structure (Figure 3c,d), on which the uniform dispersion of Pd nanoparticles provides a large surface area for oxygen access. Secondly, the carbon black prevents the wrinkled/curved surfaces of neighboring GO sheets to form enclosed spaces, facilitating oxygen transport in the catalysts [18,33]. Compared with 0.125 Pd-MFC, 0.250 Pd-MFC exhibited higher loading capacity, better conductivity, and more active sites. However, the catalyst on the electrode surface of 0.500 Pd was excessive, which affects the mass transfer of the electrode, causing the electricity output to be less than that of 0.250 Pd. These results indicate that the appropriate catalyst loading can achieve the best performance of the cathode. Therefore, we tested the cathode of the MFC in the following experiments.

### 2.3. The Electrochemical Properties of MFC

Electrochemical impedance spectroscopy was conducted for the air cathodes in the same electrolyte to determine the resistance of cathode at open circuit potentials (OCPs) [35]. To interpret the EIS results, equivalent circuits were utilized to identify the components of the cathode internal resistances. As shown in Figure 5a, it was clear that the charge transfer resistance (Rct) of the air cathodes followed the order 0.125 Pd > 0.500 Pd > 0.250 Pd. The lowest charge transfer resistance (Rct) of 0.250 Pd could be attributed to the synergy of both the reaction sites and the mass transfer at the cathode.

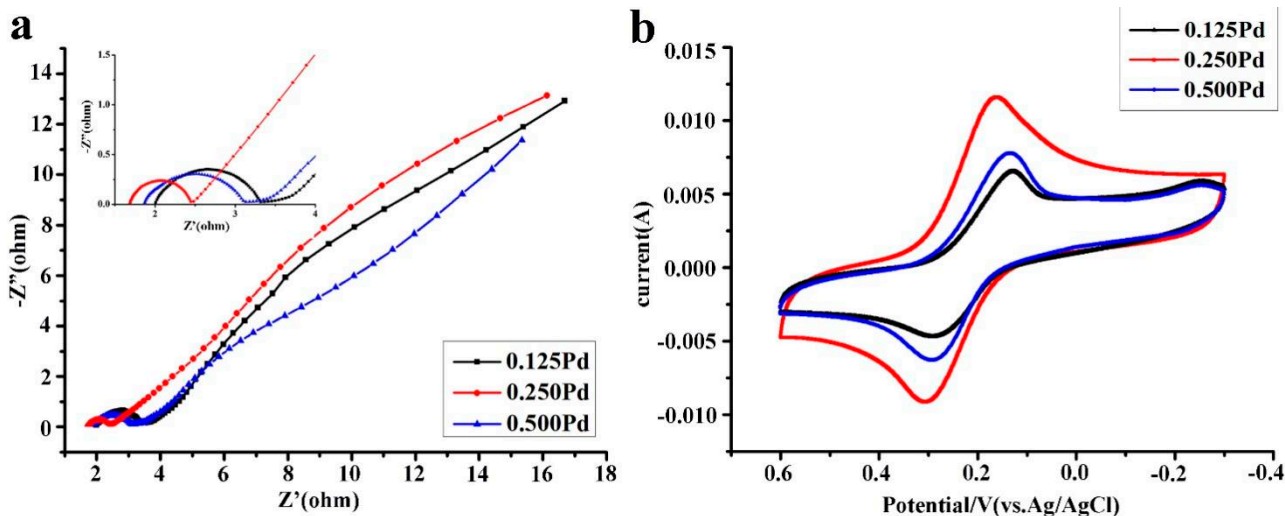

**Figure 5.** (**a**) Nyquist plots of EIS spectra of the MFCs with different Pd/GO-C cathodes. (**b**) Cyclic voltammogram of different cathodes obtained by the electrochemical workstation.

Cyclic voltammogram is the most widely used method for the electrochemical characterization of electrocatalysts, which is conducted to assess the ORR activity of different catalysts [11]. The important criterion for evaluating catalytic activities of different cathodes is peak current density. The current density increased gradually over the cycles in all the cases, indicating that cathodes with compound catalyst were very stable in the power output during long-term operation (Figure 5b). The air cathodes with 0.250 Pd generated a maximum current density of 1645 mA/m$^2$ after 11 h operation, which is higher and faster than those of the 0.125 Pd cathode (1294 mA/m$^2$ after 14 h operation) and 0.500 Pd cathode (1345 mA/m$^2$ after 14 h operation). An order of the stable state time at the maximum current density was 0.250 Pd-MFC (17 h) > 0.500 Pd-MFC (10 h) > 0.125Pd-MFC (9 h), which implied the long-term stability of the 0.250 Pd catalyst in MFC. After the MFC ran 5 cycles, the surface structure of the cathode with 0.250 Pd was scanned. The 3D spherical carbon black was assembled on the wrinkled surface of the GO; carbon black and GO overlap each other to form complex structures on the catalyst mixture surface (Figure 6).

The results of the cathodes showed in EIS (Figure 5a) and CV (Figure 5b) indicate that 0.250 mg/cm$^2$ is the optimal amount of Pd to increase the density of selective active sites for ORR without decreasing the rate of the reaction. In the air cathode, protons and electrons reach the cathode through the chamber liquid and the external circuit, respectively, and then react with oxygen to form the water [2]. The catalyst at the cathode can accept protons and electrons and adsorb oxygen to accelerate the cathode reaction [36,37]. The cathode with 0.250 Pd exhibited the best performance, which indicated that a suitable amount of catalyst plays a vital role in the efficient operation of MFC rather than that the amount of catalyst correlates with the cathodic reaction linearly. In addition, the electrochemically active surface area (ECSA) of the catalysts from the PdO reduction peak showed that the cathode with 0.250 Pd exhibited the largest ECSA than others, suggesting Pd is evenly distributed on the electrode surface [38,39]. The Pd/GO-C cathode with

different loading amounts of Pd illustrated a loose and porous agglomeration structure. However, the cathode performance was different due to the loading of Pd. This result showed that optimum special surface area and appropriate metal loading could enhance oxygen absorption and electron acceptance on the catalyst/cathode surface. In contrast, the excessive catalyst could decrease the electron transfer speed in the catalyst layer [40]. When the Pd catalyst amount was 0.250 mg/cm$^2$, the oxygen acceptability and electron transfer capacity of the cathode reached the best optimization value. The data showed that the cathode structure formed under this condition is the best with the minimal cathodic electrical conductivity and enhanced redox effect, which also indicated that the distribution of palladium, GO, and carbon black is the most uniform. Therefore, optimization of the catalyst is critical to achieving an interface with localized utilization of the active layer in the ORR.

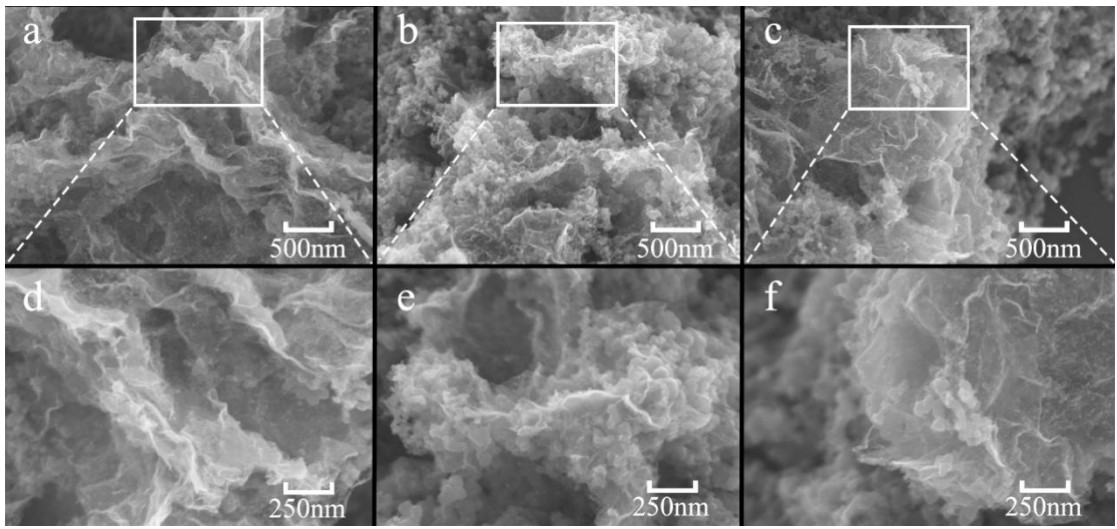

**Figure 6.** SEM images of GO-C cathode with 0.125 Pd (**a**,**d**), 0.250 Pd (**b**,**e**), and 0.500 Pd (**c**,**f**).

## 3. Methods and Materials

### 3.1. Synthesis and Characterization of the Air Cathode

The Pd/GO-C catalysts were prepared by a one-pot method as previously described [33], with a mass ratio of Pd to GO-C of 2:8 and a mass ratio of GO to carbon black of 4:6. The graphite (−325 mesh, Alfa Aesar, Ward Hill, MA, USA) was oxidized with a solution of KMnO$_4$ in aqueous H$_2$SO$_4$ and H$_3$PO$_4$ to form a liquid mixture after extended ultrasound treatment [34]. Subsequently, Vulcan carbon black (XC-72R, Cabot Co., Boston, MA, USA) and PdCl$_2$ solution (Sangon Biotech, Shanghai, China) were added to offer the ternary composites suspension. Furthermore, 5% NaBH$_4$ (Aladdin, Shanghai, China) was used to reduce Pd after adjusting pH to 10. The suspension was filtered, rinsed with deionized water, and dried under vacuum at 50 °C for 10 h to afford Pd/GO-C catalyst powder. The Pd/GO-C catalyst was mixed with 50% ethanol to afford 10 mg/mL suspension of catalysts, which was sonicated for 30 min to complete the dispersion. The suspensions with volumes of 0.75 mL, 1.5 mL, and 3 mL were dripped onto the surface of the 12 cm$^2$ cathode carbon papers (CP, Toray Carbon Paper, TGP-H-60, Alfa Aesar, Ward Hill, MA, USA) with 0.15 mL suspension for each layer, respectively. Each layer was dropped after previous layer was completely dried; then, 5% Nafion (5% *w/w*, DuPont, Wilmington, DE, USA) solution was dropped to the electrode surface with 1/5 volume of the suspension. In this process, different amount of Pd/GO-C catalysts were covered on the cathode to form the different Pd loadings (0.125 mg/cm$^2$, 0.250 mg/cm$^2$, and 0.500 mg/cm$^2$).

### 3.2. MFC Construction and Operation

In this study, the single chamber microbial fuel cell (SCMFC) electrode chamber was a cylindrical closed structure with an effective volume of 28 mL. The cathode was directly exposed to air. The cathode side contacting with the solution was adhered to the catalysts, while the air side of the cathode consisted of the carbon base layer and PTFE (Poly Tetra Fluoroethylene) diffusion layer (Figure 2). The $2 \times 2$ cm$^2$ carbon felt (CF, Fuel Cell Earth, AvCarb® C series PAN-based soft carbon felts, 3.2 mm, Boston, MA, USA) was used as the anode being pretreated by the following procedures. The CF was soaked in 1 mol/L HCl and 1 mol/L NaOH solution for 2 h, separately, to remove grease and impurity ions on the surface of CF, and then, it was repeatedly soaked in deionized water until the pH reached 7. The SCMFC was inoculated with anaerobic sludge (obtained from Jinan western sewage treatment plant, Jinan, China) to form an active biofilm. M9 buffer was used in an anode chamber, which contained $NH_4Cl$ (1 g/L), NaCl (0.5 g/L), $Na_2HPO_4 \cdot 12H_2O$ (17.8 g/L), $CaCl_2$ (0.11 g/L), $MgSO_4$ (0.247 g/L), $KH_2PO_4$ (3 g/L), and sodium acetate (20 mmol/L) as a carbon source. Both the electrodes were connected with the copper wire and external resistance (1 K$\Omega$). The device was incubated at 30 °C and connected to a data acquisition system (MPS-010602, QichuangMofei Co., Beijing, China) to collect the output voltage of the SCMFCs from different cathodes (per minute). According to the operation cycle of SCMFCs, the anode solution was refreshed every two days, and the MFC device was operated in batch mode. The current was obtained by dividing the voltage by the resistance, and the current density was obtained by dividing the current by the projected area of the cathode. Some of the data obtained in this paper were compared with the traditional catalyst Pt/C. The Pt/C paper (Shanghai Hesen Co., Ltd., Shanghai, China) with a platinum content of 0.5 mg/cm$^2$ was used in the cathode chamber.

### 3.3. Characterization and Electrochemical Measurements

According to the output voltage acquired by the computer, current density (mA/m$^2$) was calculated from U/R/A, and power density (mW/m$^2$) was calculated from U·I/A, wherein U, R, I, and A represent voltage, resistance, current, and area, respectively. To obtain the MFC maximum output power density, the external resistance with a resistor box (J-2361) varied from 10 to 0.1 k$\Omega$ until it reached a steady state, and the polarization curve was obtained from the corresponding voltage. Electrochemical analyses such as CV and EIS were performed by using a CHI 660E electrochemical workstation (CH Instruments, Austin, USA). Field SEM (Quanta FEG 250, FEI, USA) was used to evaluate if the composite catalyst layers were evenly dispersed on the surface of the working cathode as well as the structure and morphology of composite catalysts. The morphology of catalyst surface was analyzed with EDS (Bruker Nano GmbH, Berlin, Germany). XPS (Axis Ultra DLD spectrometer, Kratos Analytical Ltd., Stretford, UK) was used to measure the elemental composition and chemical valence of the catalysts with a monochromatized Al K$\alpha$ X-ray source, and the binding energies were calibrated using C 1s peak at 283.41 eV and Pd 3d peak at 347.03 eV. In this study, the three-electrode cell consisted of an Ag/AgCl (+197 mV vs. SHE, saturated KCl) electrode was used as a reference electrode with the anode CF serving as a counter electrode and the cathode covered with different catalysts serving as a working electrode. CV analysis was performed from $-0.3$ to $+0.6$ V with a scan rate of 10 mV/s. EIS for the composite cathode was performed in a frequency range of 100–0.01 Hz, and the initial voltage was set according to the open-circuit voltage (OCV) of MFCs. Surface morphology images of working cathode were taken by SEM operating at 5 kV below.

The morphology of the sample was assessed by TEM (JEM-2100, JEOL, Japan) at 200 kV accelerating voltage to observe the Pd/GO-C catalyst. High-resolution transmission electron microscopy (HR-TEM) was carried out with a JEM-2100 (JEOL, Japan).

### 4. Conclusions

In summary, evenly distributed palladium on the surface of GO-C improved the ORR of the air cathode and the performance of air cathode MFC significantly. Appropriate

metal loading also substantially affected the catalytic efficiency of the cathode catalyst. The cathode doped with suitable palladium (0.250 mg/cm$^2$) catalyst exhibited the highest current density (1645 mA/m$^2$) and lowest Rct, which not only increase the oxygen reduction activity but also improve the electromotive force. The MFC employing the prepared cathode catalyst afforded the highest output voltage of 658 mV and the maximum power density of 901 mW/m$^2$. This study provides an innovative way for the preparation of a high-efficient ORR catalyst for MFCs.

**Supplementary Materials:** The following are available online at https://www.mdpi.com/article/10.3390/catal11080888/s1, Figure S1: TEM (a) and HR-TEM (b) images of Pd/GO-C cathode catalyst.

**Author Contributions:** J.W.: Conceptualization, Methodology, Writing—original draft. K.M.: Methodology, Software, Validation. X.Z.: Data curation, Methodology, Software. D.L.: Data curation, Methodology, Software. X.Y.: Methodology, Software. W.L.: Methodology, Writing—review and editing. J.C.: Writing—review and editing. J.Y.: Investigation, Software. Q.Y.: Writing—review and editing, Conceptualization, Methodology Software. All authors have read and agreed to the published version of the manuscript.

**Funding:** The authors are grateful for the financial support from the National Natural Science Foundation of China (Grant No. 31800116), Natural Science Foundation of Shandong Province, China (Grant No. ZR2018LC004), Young Doctors Cooperation Fund of Qilu University of Technology, Shandong Academy of Sciences (Grant No. 2017BSHZ025), Key Technology Research and Development Program of Shandong Province (2019GSF111014), Introduction and Cultivation Plan of Young Innovative Talents in Colleges and Universities of Shandong Province, and National Innovation and Entrepreneurship Training Program for Local College Student (No. S202010431030).

**Data Availability Statement:** The data presented in this study are available on request from the corresponding author. The data are not publicly available due to the need of follow-up research.

**Conflicts of Interest:** The authors declare no conflict of interest.

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
