# Peer review of "Uniform Distribution of Pd on GO-C Catalysts for Enhancing the Performance of Air Cathode Microbial Fuel Cell"

_catalysts, doi:10.3390/catal11080888_

Round 1

Reviewer 1 Report

In this study, the uniform distribution of Pd on GO-C catalysts for enhancing the performance of air-cathode microbial fuel cell was investigated. This seems to be an interesting topic. However, the following issues need to be revised before this article is published:

  1. Figure 2 is unreasonable. Why do the authors draw two schematic diagrams of GO, and then after GO and black carbon are mixed, does black carbon really bonding with the two pieces of GO?
  2. What does the yellow ball mean in Figure 2?
  3. Part 3 (Methods and Materials) should be put before Part 2 (Results and discussion).
  4. In the section of Methods and Materials, the author should make a detailed description of the place where the experimental materials were purchased. Such as 2.1 Materials

Author Response

Comment # 1: Figure 2 is unreasonable. Why do the authors draw two schematic diagrams of GO, and then after GO and black carbon are mixed, does black carbon really bonding with the two pieces of GO?

Thank you for the helpful suggestions. In the new version, in order to achieve better results, we modified the pictures by increasing the number of graphene and changing the angle of the image. Through the process in the picture, we finally obtain the graphene layers separated by carbon black, and Pd is evenly dispersed in them.

Comment # 2: What does the yellow ball mean in Figure 2?

Thank you for careful reading! The yellow ball represents Pd. We have marked it in Figure 2 according to your opinion.

Comment # 3: Part 3 (Methods and Materials) should be put before Part 2 (Results and discussion).

Thank you for your suggestion. In the new revision, we put the Part 3 (Methods and Materials) before the Part 2.

Comment # 4: In the section of Methods and Materials, the author should make a detailed description of the place where the experimental materials were purchased. Such as 2.1 Materials

Thank you. The purchasing channels of the materials were annotated in Methods and Materials.

Reviewer 2 Report

The paper presents a study regarding uniform distribution of Pd on GO-C catalysts for enhancing the performance of air-cathode microbial fuel cell. I considered that the topic of the paper is very interesting and report new information. However, I consider that the paper has to be improved in order to be published in Catalysts, and my comments are presented bellow:  
- My first observation concerns the structure of the paper, in the sense of placing the Results and discussion before Methods and Materials that creat confusion and difficulty of reading the paper. 

- In the Results and discussions there are no data about each element concentration (C, Pd and N2) that can be obtained from XPS and EDS measurements. Therefore I suggest the authors to mention in the text or in a table these data.  Moreover, when these characterization methods are used, which solution described in the Methods and Materials was analyzed?

Author Response

Comment # 1: My first observation concerns the structure of the paper, in the sense of placing the Results and discussion before Methods and Materials that creat confusion and difficulty of reading the paper. 

Thank you. According to your suggestion, we adjusted the order of the two parts in the new revision.

Comment # 2: In the Results and discussions there are no data about each element concentration (C, Pd and N2) that can be obtained from XPS and EDS measurements. Therefore I suggest the authors to mention in the text or in a table these data. Moreover, when these characterization methods are used, which solution described in the Methods and Materials was analyzed?

Thank you for the helpful suggestions. The relevant data were described in the text, the element concentration of C, N and Pd were 44.25%, 34.51% and 21.24%. In line 138 of the article, the substances determined by EDS and XPS were added.